

# Predicting disease occurrence of cabbage Verticillium wilt in monoculture using species distribution modeling

Kentaro Ikeda[1] and Takeshi Osawa[2]

[1] Department of Agriculture, Gunma Prefectural Office, Isesaki, Gunma, Japan
[2] Graduate School of Urban Environmental Sciences, Tokyo Metropolitan University, Hachioji, Tokyo, Japan

## ABSTRACT

**Background:** Although integrated pest management (IPM) is essential for conservation agriculture, this method can be inadequate for severely infected fields. The ability to predict the potential occurrence of severe infestation of soil-borne disease would enable farmers to adopt suitable methods for high-risk areas, such as soil disinfestation, and apply other options for lower risk areas. Recently, researchers have used species distribution modeling (SDM) to predict the occurrence of target plant and animal species based on various environmental variables. In this study, we applied this technique to predict and map the occurrence probability of a soil-borne disease, Verticillium wilt, using cabbage as a case study.

**Methods:** A disease survey assessing the distribution of Verticillium wilt in cabbage fields in Tsumagoi village (central Honshu, Japan) was conducted two or three times annually from 1997 to 2013. Road density, elevation and topographic wetness index (TWI) were selected as explanatory variables for disease occurrence potential. A model of occurrence probability of Verticillium wilt was constructed using the MaxEnt software for SDM analysis. As the disease survey was mainly conducted in an agricultural area, the area was weighted as "Bias Grid" and area except for the agricultural area was set as background.

**Results:** Grids with disease occurrence showed a high degree of coincidence with those with a high probability occurrence. The highest contribution to the prediction of disease occurrence was the variable *road density* at 97.1%, followed by *TWI* at 2.3%, and *elevation* at 0.5%. The highest permutation importance was *road density* at 93.0%, followed by *TWI* at 7.0%, while the variable *elevation* at 0.0%. This method of predicting disease probability occurrence can help with disease monitoring in areas with high probability occurrence and inform farmers about the selection of control measures.

Corresponding author
Kentaro Ikeda,
ikeda-ken@pref.gunma.lg.jp

## INTRODUCTION

Intensive agriculture has supported world food production since World War II (*Moffatt, 2020*). However, the sustainability of intensive agriculture is often questioned because of negative consequences such as soil and water pollution, risks to human health from

pesticides and fertilizer, and water resource depletion by over-exploitation of water resources (*Oosterbaan, 1989*; *Matson et al., 1997*; *Tilman et al., 2002*; *Hernández et al., 2006*). To address these problems, a shift to conservation agriculture is required (*Reicosky, 2003*; *Kassam et al., 2009*). On the other hand, measures to increase food supply to maintain food security are also needed because the world population is expected to reach over 9.7 billion by 2050 (*United Nations, 2015*). These competing demands for sustainability and higher yields presents a complex problem facing modern intensive agriculture (*Brussaard et al., 2010*), but it is a challenge that must be met.

To cope with the ever growing global food demand, intensive agriculture primarily revolves around large-scale monoculture, the continuous or consecutive growth of the same crop over large areas (*Cook & Weller, 2004*). Monocultures carry a heavy risk of soil-borne disease because pathogens have a continuous supply of host plant and are thus able to persist or even accumulate in the soil (*Newton, Begg & Swanston, 2009*; *Jenking & Jain, 2010*). Soil disinfestation by fumigation with chemical control agents can be effective in the management of soil-borne disease (*Wilhelm & Paulus, 1980*; *Koike et al., 2003*), but also carries the risk of negative impacts on the environment and human health (*Hernández et al., 2006*; *Sande et al., 2011*; *United States Environmental Protection Agency, 2017*). In addition, as more fields require chemical control, the costs associated therewith also increase (*Landis, 1987*; *Labrada & Fornasari, 2001*; *Koike, Gladders & Paulus, 2006*). So, although effective, chemical disinfestation to reduce soil-borne disease can be difficult to adopt when the costs and consequences outweigh the benefits.

Integrated pest management (IPM), which is defined as the long-term prevention of pests or their damage through a combination of techniques, is essential for sustainable agriculture (*Apple, Ray & Smith, 1976*; *UC IPM, 2015*). The control of soil-borne disease in IPM typically involves the application of resistant cultivars, crop rotation, exclusion and prevention of the pathogen's inoculum source (*Tsushima, 2014*; *Ikeda et al., 2015*). However, such methods are inadequate to bring about control in severely infested fields. Therefore, intensive continuous soil fumigation may be justified in severely infested fields (*Krikun, Netzer & Sofer, 1974*). If the potential occurrence of severe infection of soil-borne disease could be predicted, farmers could adopt soil fumigation for high-risk areas and apply other options for areas of lower risk. Furthermore, evaluating infestation risk in a given area can help growers to prevent the spread to unaffected fields or re-introduction in previously managed fields. This approach is in accordance with the IPM framework for sustainable, low-environmental impact, and cost-effective agriculture (*Tsushima & Yoshida, 2012*). Control measures for soil-borne diseases should be selected according to the disease potential occurrence of each target area. However, the development of a practical method to predict such disease potential occurrence has not previously been given in the field of plant pathology.

Ecological science uses species distribution modeling (SDM) (*Osawa et al., 2011*; *Osawa, 2015*) for predicting the probability of occurrence and sometimes also abundance of target plant and animal species based on environmental variables such as terrain and climate factors (*Guisan & Zimmermann, 2000*; *Soberón & Peterson, 2005*; *Soberón, 2007*; *Peterson et al., 2011*; *Peterson, 2014*). SDM can predict the distribution of a target

species using survey data obtained in the area (*McCune, 2016*). Additionally, the respective contribution, or importance, of environmental variables on species distribution can be calculated. Maximum entropy algorithm is one of the most famous SDM approaches and can be used to predict species potential distribution by analyzing presence-only data with environmental variables (*Phillips, Dudik & Schapire, 2004*; *Phillips, Anderson & Schapire, 2006*). MaxEnt software has been successfully used in predicting potential species distribution (*Ortega-Huerta & Peterson, 2008*). Recently, MaxEnt has been used to predict plant disease distribution, such as Fusarium dry root rot in common beans, *Phomopsis vaccinii* in Vaccinium species, myrtle rust in Myrtaceae family and *Pseudomonas syringae* pv. *actinidiae* in kiwifruit (*Cunniffe et al., 2016*; *Macedo et al., 2017*; *Narouei-Khandan et al., 2017*; *Berthon et al., 2018*; *Wang et al., 2018*; *Narouei-Khandan et al., 2020*). In the present study, we used MaxEnt to predict the occurrence probability of Verticillium wilt, a soil-borne disease, as a case study on cabbage field in Japan.

Tsumagoi village in the Gunma prefecture of Japan intensively produces cabbage as one of the main production sites for this crop in Japan. This is an ideal site to test the application of the SDM technique, given its large-scale area of consolidated monoculture and years of field monitoring, which enable a detailed analysis of the occurrence of Verticillium wilt. Verticillium wilt in cabbage has been a big problem in this region's cabbage production since 1994 (*Shiraishi et al., 2000*). This is a soil-borne disease caused by *Verticillium dahliae* and *V. longisporum* (*Banno et al., 2011*). The cabbage farmers in Tsumagoi village have suffered severe economic losses from this disease (*Shiraishi et al., 2000*).

The objective of the study was to construct an occurrence probability map of Verticillium wilt of cabbage in Tsumagoi village in Gunma prefecture in Japan to test the application of SDM to soil-borne plant disease. The findings aim to inform farmers to choose adequate control measures, such as introducing resistant cultivars or changing cultivation period. This trial can contribute to the development of IPM by offering both environmental and economic improvements to soil-borne disease management in agricultural production.

## MATERIALS AND METHODS

### Study area

The distribution of Verticillium wilt of cabbage was surveyed in Tsumagoi village (36°51′ 17″N, 138°53′00″E) in central Honshu, the largest of Japan's four main islands (Fig. 1). The climate is subarctic humid climate (Dfb) in Köppen climate classification with an elevation ranging between 700 and 1,500 m above sea level and a mean annual temperature of 7.2 °C (*Japan Meteorological Agency, 2017*; *Village Office of Tsumagoi, 2017*). Cabbages are produced on approximately 2,790 ha of the 3,200 ha agricultural area. This is a remarkably large monoculture site for cabbage production in Japan. Thus, the village is renowned in Japan as a production site for cabbage. Cabbages are harvested there from April (early spring) to October (autumn) (*Village Office of Tsumagoi, 2017*). However,

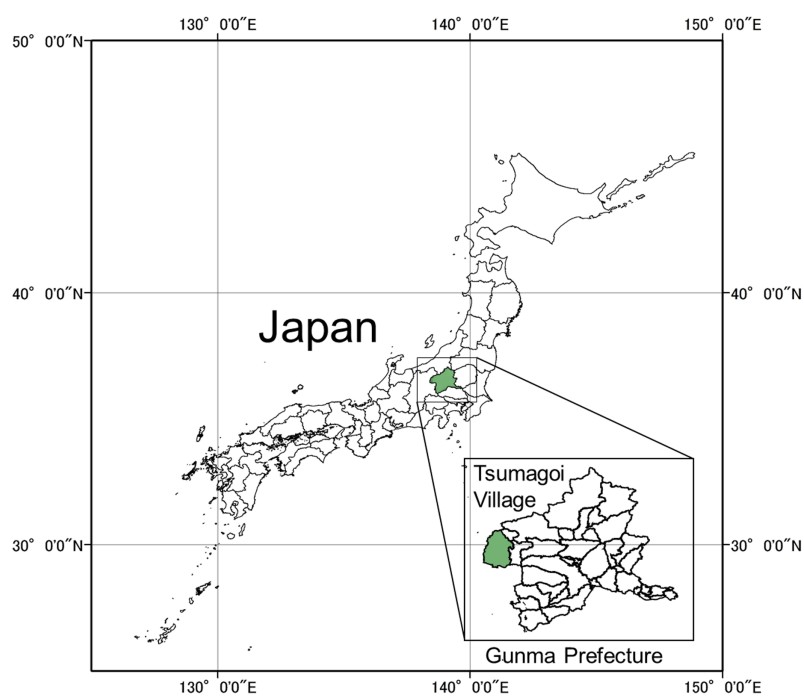

**Figure 1 Location of study area.** This study was conducted in Tsumagoi village (36°51′17″N, 138°53′00″E) of Gunma Prefecture in central Honshu, the largest of Japan's four main islands.

potato, maize and scarlet runner bean are also grown in this area, in significantly lower proportions than those of cabbage (*Village Office of Tsumagoi, 2017*).

## Target disease and disease survey

Cabbage production in the area has suffered from Verticillium wilt since 1994 (*Shiraishi et al., 2000*). Verticillium wilt is a soil-borne disease associated with *Verticillium dahliae* and *V. longisporum* (*Banno et al., 2011*), which can survive for years using microsclerotia as a resting structure. *V. dahliae* has a board range of hosts, mainly plants from the Solanaceae and Brassicaceae families (*Pegg & Brady, 2002*), while *V. longisporum* infects mainly members of the Brassicaceae family (*Inderbitzin et al., 2013*). In our study site, *V. dahliae* was found to be the dominant species causing Vertillium wilt in cabbage (*Banno et al., 2015*). In recent years, cultivars resistant to Verticillium wilt have been planted in the village. The disease survey was conducted two or three times per year annually from 1997 to 2013, 38 times in total by official and private agricultural extension workers and plant pathologists capable of distinguishing the disease in all fields of the village. This survey and field collections were approved by Gunma Prefectural Office under permission document no. H28.114.30. Disease occurrence was identified by external symptoms such as outer leaf yellowing and wilting (Figs. 2A and 2B). When a cabbage suspected to be infected was found, the observers examined the vascular system of selected cabbages to identify browning and thus confirm the existence of the disease (Fig. 2C). The locations of fields where the disease occurred in field survey were marked on a 1:5,000 map. When new records were found, investigators took cabbage samples for

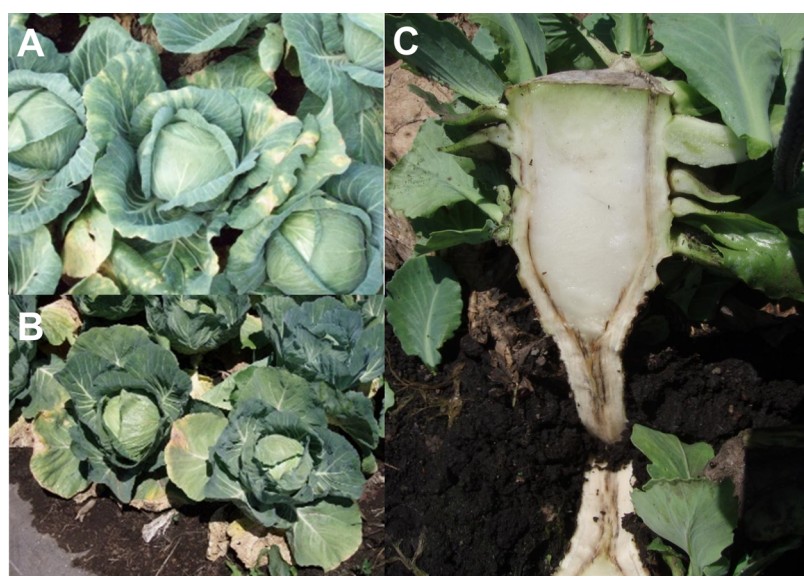

**Figure 2 Symptoms of cabbage Verticillium wilt.** (A & B) Leaves wilting and yellowing. (C) Browning vascular systems.

isolation and identification of the pathogens. The study area was divided into a grid of 500 × 500 m cells. The grid cell size was selected according to both the degree of accuracy of the disease survey and the scale of land ownership. Disease occurrence (a binary variable) of both pathogen species and the values of the selected explanatory variables were recorded for each grid cell (Figs. 3A and 3B).

## Explanatory variables

Geographic Information System software ArcMap (ver. 10.7.1; ESRI, Redlands, CA, USA) was used for analysis of the data set. We selected road density, elevation and topographic wetness index (TWI) as possible explanatory variables that may affect the disease potential occurrence. In Tsumagoi village, the pathogens of cabbage Vertillium wilt were detected in field soil (*Banno et al., 2011*) and there was serious soil erosion from cabbage fields to the roads (*Deb, 2006*). This suggested to us that the disease spread from one field to another by pathogen-infested soil, which prompted us to investigated road density as an explanatory variables. Furthermore, we included elevation as an explanatory variables because of relationship between elevation and temperature: for every 100 m increase in elevation, temperature rises by about 0.6 °C. Finally, we also selected TWI, which indicates the degree of soil moisture (*Gallant, 2000*), the latter is known to influence the development of soil-borne disease like Verticillium wilt (*Pegg & Brady, 2002*).
Road data were downloaded as polyline data from Fundamental Geospatial Data (Geographical Survey Institute; http://www.gsi.go.jp/kiban/, accessed on 24 June 2020). The number of new roads had not increased dramatically from 2013 to 2016 in the study area. The total length of road in each grid cell was calculated from the polyline data as "road density." Elevation and slope were downloaded from the National Land Numerical Information download service (Ministry of Land, Infrastructure, Transport and Tourism,
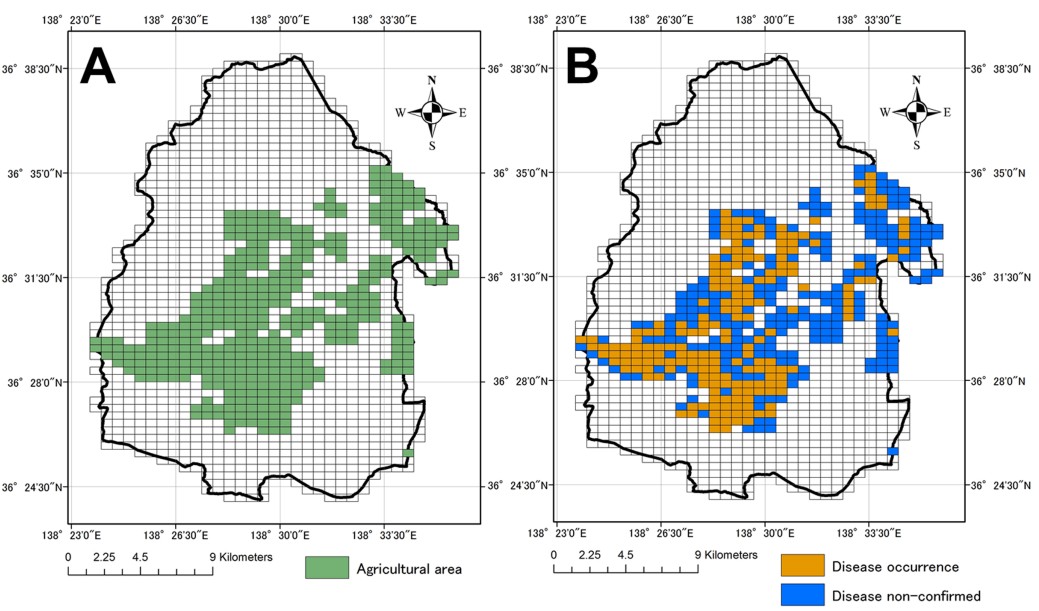

**Figure 3 Agricultural area in study area and disease occurrence.** Each grid cell is 500 × 500 m. (A) agricultural area in Tsumagoi village, (B) occurrence of Verticillium wilt of cabbage in grid cells Base maps, including grid map, and distribution data of agricultural area were obtained from Fundamental Geospatial Data (Geographical Survey Institute; http://www.gsi.go.jp/kiban/, accessed on 6 January 2016) and National Land Numerical Information download service (Ministry of Land, Infrastructure, Transport and Tourism, http://nlftp.mlit.go.jp/ksj/, accessed on 6 January 2016), respectively.

http://nlftp.mlit.go.jp/ksj/, accessed on 24 June 2020). Slope data are required to calculate TWI (*Gallant, 2000*), which is defined as follows:

$$\text{TWI} = \ln \frac{a}{\tan b}$$

where $a$ is local catchment area (i.e., the local upslope area draining per unit) and $b$ is the local slope. Values for elevation and TWI were applied to each grid cell along with the road density variable. The agricultural area shown in Fig. 3A were also downloaded from the National Land Numerical Information download service described above.

## Model construction

The model of probability of disease occurrence was constructed using SDM software MaxEnt ver. 3.3.3k (downloaded at: https://biodiversityinformatics.amnh.org/open_ source/maxent/) (*Phillips, Dudik & Schapire, 2004*; *Phillips, Anderson & Schapire, 2006*). This software was chosen as it can analyze presence-only data efficiently with a low number of occurrences (*Elith et al., 2006*). MaxEnt is used for predicting habitat suitability of target species, but here we applied it to the prediction of probability of soil-borne disease occurrence. The disease occurrence probability (0–1) in each grid cell, percent contribution and permutation importance of each explanatory variables were calculated by MaxEnt. We conducted the analysis using the whole village area, given that this was the scale relevant for our study. However, the original disease survey was mainly

conducted within the agricultural areas (Fig. 3A), hence we used a "Bias Grid" weighting and set the remaining "non-agricultural area" as background in MaxEnt setting. To avoid overfitting, only "Liner" and "Quadractic" functions were used and the value for "Regularization multiplier" was changed from "1" to "2" in the MaxEnt settings (*Merow, Smith & Silander, 2013*; *Radosavljevic & Anderson, 2014*; *Syfert, Smith & Coomes, 2013*). Cross-validation was selected in "Replicated run type" and repeated 20 times. The model of the probability of disease occurrence was also estimated by means of receiver operating characteristic (ROC) analysis. Furthermore, correctly classified instances (CCI), sensitivity, specificity, true skill statistics (TSS) were calculated based on the "threshold by maximum training sensitivity plus specificity" in MaxEnt to evaluate model performance.

## RESULTS

There were 1,392 grids cells in our study area, we confirmed disease occurrence in 194 grid cells (Fig. 3B). The SDM model constructed with MaxEnt was used to map the probability of disease occurrence and points of disease occurrence (Fig. 4). Threshold by Maximum training sensitivity plus specificity from MaxEnt analysis was 0.543. The disease occurrence probability map was created based on this threshold. Grids in this map were colored when the grids had a higher probability compared to the threshold (in total, 556/1,392 grids in this study area). The grids with disease occurrence had a high degree of coincidence with those with a high probability of such occurrence.

Percent contribution and permutation importance of predictor variables are shown in Table 1. The highest contribution to the prediction was road density at 97.1%, followed by TWI at 2.3%, and elevation at 0.5%. Permutation importance was as follow: road density at 93.0%, TWI at 7.0%, and elevation at 0.0%. Both percent contribution and permutation importance indicated that road density was the best predictor variable: disease probability occurrence was highest when road density was about 4,000 m/grid (Fig. 5).

Estimation indices for accuracy of MaxEnt modeling of cabbage Verticillium wilt is shown in Table 2. The constructed model represented the occurrence of Verticillium wilt of cabbage moderately well, as demonstrated by the value, 0.861, of the AUC average of the ROC. The calculated values for CCI, sensitivity, specificity, and TSS using the threshold were 0.705, 0.876, 0.678 and 0.544, respectively.

## DISCUSSION

In this study, we applied SDM to predict the occurrence probability of Verticillium wilt in cabbage as a case study. Our model could provide the occurrence of Verticillium wilt of cabbage in Tsumagoi village to some degree. To the best of our knowledge, this is the first time SDM has been used to estimate occurrence probability of soil-borne disease. Our findings indicated that this method can successfully predict disease occurrence probability using survey data and selected environmental variables in a monoculture area.

Our result showed that road density significantly influenced disease occurrence probability (Table 1), with the highest probability of disease at around 4,000 m/grid

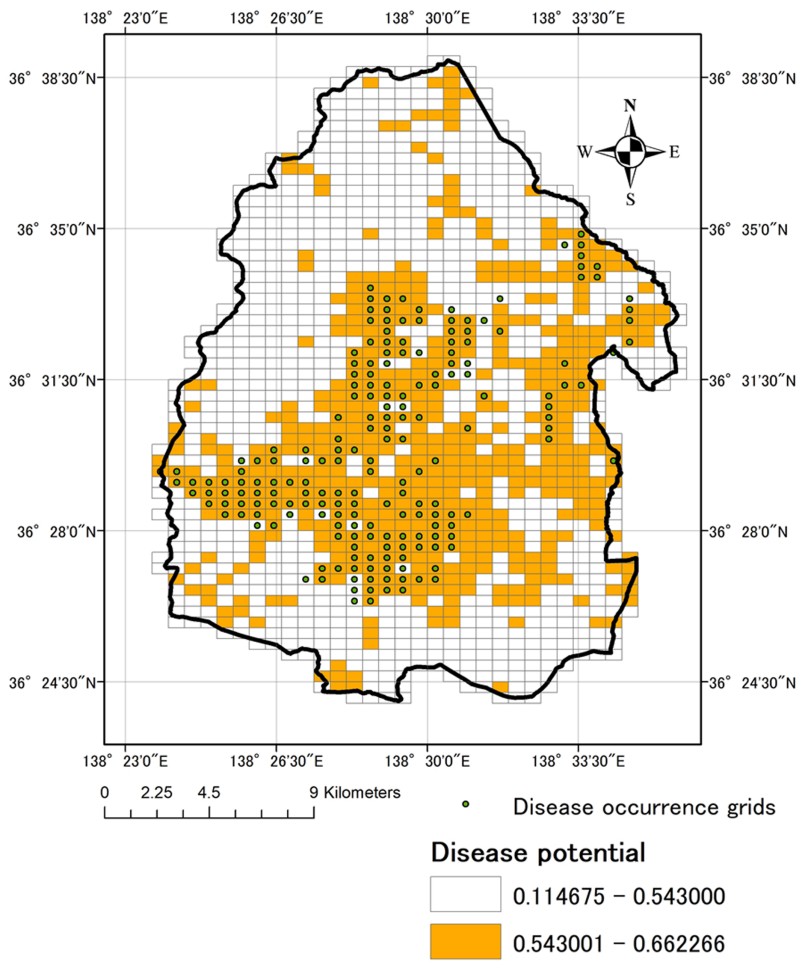

**Figure 4 Disease occurrence grids and Distribution of occurrence probability of Verticillium wilt.**
Each grid cell is 500 × 500 m. The occurrence probability in each grid was calculated based on disease occurrence data and Three explanatory variables (road density, elevation and topological wetness index) using by MaxEnt Ver. 3.3.3k (downloaded at https://www.cs.princeton.edu/~schapire/maxent/).

**Table 1 Percent contribution and permutation importance of various variables in MaxEnt modering of cabbage Verticillium wilt.**

| Variables | Percent contribution | Permutation importance |
|---|---|---|
| Road (m) | 97.1 | 93.0 |
| TWI | 2.3 | 7.0 |
| Elevation (m) | 0.5 | 0.0 |

(Fig. 5). *Numminen & Laine (2020)* showed that road networks played an important role in the spread of plant diseases, which is consistent with our findings, suggested that the pathogen spread via road networks. It was thought that the pathogen was locally transmitted between fields through the soil, as they are soil-borne (*Banno et al., 2011*).

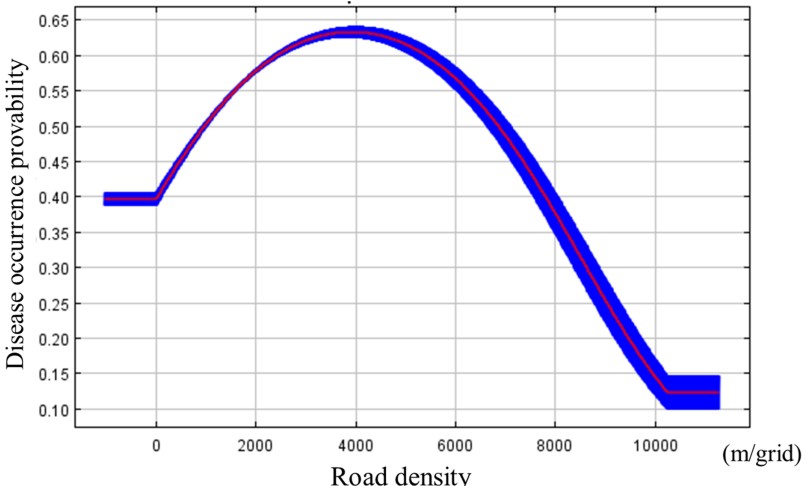

**Figure 5 Marginal response curve between the disease potential occurrence calculated by MaxEnt and road density.** Blue band shows standard deviation.

**Table 2 Evaluation indices of estimation accuracy in MaxEnt modeling of cabbage Verticillium wilt.**

| Evaluation indices | Values |
| --- | --- |
| Area Under Curve (AUC) | 0.861 |
| Threshold by Maximum training sensitivity plus specificity | 0.543 |
| Correctly Classified Instances (CCI) | 0.705 |
| Sensitivity | 0.876 |
| Specificity | 0.678 |
| True Skill Statistics (TSS) | 0.554 |

Soil can be transferred from infested fields to road surfaces on the wheels of tractors or other vehicles and even the footwear of farmworkers. Additionally, Tsumagoi village is located in mountainous area with steep slopes, which facilitates soil transfer between fields (e.g., by wind and water). Both natural and human-mediated processes can influence the spread of Verticillium wilt in our study area.

On the other hand, our model suggested that the influence of TWI was quite low, even though soil moisture has previously been known to influence the development of soil-borne disease like Verticillium wilt (*Pegg & Brady, 2002*). The relationship between Verticillium wilt in cabbage and soil moisture had not yet been investigated before this study; however, it was reported that soil moisture was not significant factor for disease development in oilseed rape, Brassicaceae family (*Knüfer, 2013*). Consequently, the low impact shown by TWI on disease occurrence probability concurs with previous research, although more research is required to conclusively rule out this relationship. Elevation also showed a very low contribution to the probability of disease occurrence. In general, there are a correlation between elevation and temperature, as temperature rises about 0.6 °C for every 100 m increase in elevation. Given that the optimal temperature for

cabbage Verticillium wilt was between 19 °C and 23 °C, temperature during the harvest period is regarded as an especially important (*Kemmochi et al., 2001*). Therefore, we predicted that elevation could have an influence on disease occurrence. Cabbage is normally cultivated between 4 °C and 24 °C (*Bradley & Courtier, 2006*). Cabbage cultivation period was determined according to the elevation of each field because temperature decreases with increasing elevation. Given our results, the elevation factor might not influence the probability of disease occurrence in our study area.

Our model was evaluated using statistic metrics based on the defined threshold (Table 2) and the disease occurrence probability map (Fig. 4). It appeared that many grids where disease occurrence was recorded had a high probability. However, the AUC, CCI, sensitivity, specificity and TSS obtained from our analysis were moderate compared with other studies (*Wang et al., 2018*; *Narouei-Khandan et al., 2020*). The sensitivity of the model should be regarded as the most important for our objective because we hope use it to monitor grids with a high disease occurrence probability. The sensitivity of the constructed model was 0.88, which indicates that the model can predict the disease occurrence grids at about 88% accuracy. As a result, the model constructed using MaxEnt was sufficiently effective for prediction of the disease occurrence.

The disease occurrence probability map helps disease monitoring in high probability grids and selected area, and can inform the application of measures such as fumigation, planting disease-resistant cultivars, and changing the timing of cabbage cultivation (*Kemmochi et al., 2001*). Although it is not possible to change the road density, it is possible to apply soil erosion and sediment transport control technologies such as the establishment of cover crops (*Lawson et al., 2015*) to decrease the potential occurrence of the disease, especially in those grid cells where there are preexisting records and those grid cells predicted to have high occurrence probability. In addition, we recommend that the farmers and extension workers apply methods to reduce potential spread of infested soil, such as using portable vehicle-washing equipment and changing footwear at the site are required.

This study shows that SDM analysis using MaxEnt can be used to predict the occurrence using survey data and environmental variables. This approach could be applied to other intensive crop cultivation areas with accurate disease survey data and environmental variables, available as GIS data, that potentially influence disease occurrence.

## CONCLUSIONS

In this study, we applied SDM to predict the occurrence probability of Verticillium wilt of cabbage, soil-borne plant disease. The SDM model constructed from three explanatory variables provided prediction of the occurrence of Verticillium wilt of cabbage in Tsumagoi village, Japan. The model developed from the field survey data showed that road density played an important role in the occurrence of the disease. Key points of this study are; (1) SDM enabled us to predict the probability of disease occurrence in a relatively narrow area; (2) potential explanatory variables for disease occurrence were predicted

based on SDM using field survey; and (3) predicting the probability of disease occurrence can help farmers to select suitable control methods for high and low-risk areas.

## ACKNOWLEDGEMENTS

We are grateful to Shigenobu Yoshida, Seiya Tsushima, Hiroshi Sakai, Toshihiko Urushibara, Toshimasa Shiraishi, and extension workers of Japan Agricultural Corporative of Tsumagoi for their useful advice and technical support.

### Funding

This work was supported by a grant from the Science and Technology Research Promotion Program for Agriculture, Forestry, Fisheries and Food Industry (25056C) from the Ministry of Agriculture, Forestry and Fisheries of Japan, Gunma New Agricultural Frontier Fund from Gunma Prefectural Government of Japan, and JSPS KAKENHI Grant Number 16H05061. There was no additional external funding received for this study. The funders had no role in study design, data collection and analysis, decision to publish, or preparation of the manuscript.

### Grant Disclosures

The following grant information was disclosed by the authors:
Ministry of Agriculture, Forestry and Fisheries of Japan.
Gunma New Agricultural Frontier Fund from Gunma Prefectural Government of Japan.
JSPS KAKENHI: 16H05061.

### Competing Interests

The authors declare that they have no competing interests.

### Author Contributions

- Kentaro Ikeda conceived and designed the experiments, performed the experiments, analyzed the data, prepared figures and/or tables, authored or reviewed drafts of the paper, and approved the final draft.
- Takeshi Osawa conceived and designed the experiments, analyzed the data, authored or reviewed drafts of the paper, and approved the final draft.

### Field Study Permissions

The following information was supplied relating to field study approvals (i.e., approving body and any reference numbers):

Disease surveys were approved by Gunma prefecture Office (permission document no.: H28.114.30).

### Data Availability

The raw measurements are available as Supplemental Files.

## Supplemental Information

Supplemental information for this article can be found online at http://dx.doi.org/10.7717/peerj.10290#supplemental-information.

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
