# Peer review of "Predicting disease occurrence of cabbage Verticillium wilt in monoculture using species distribution modeling"

_PeerJ, doi:10.7717/peerj.10290_

## Round 0.1 · original submission · Major Revisions

The four reviewers that checked your manuscript had conflicting opinions on your paper. I suggest that you consider all of them. Many of the issues raised by reviewers were in relation to methods and suggested additional analyses for improving results. Others are related to sampling and more detail is needed to understand results. I think this is novel research that uses SDMs to predict disease occurrence, thus I recommend taking carefully every issue by reviewers.

Reviewer 1 ·

Basic reporting

The manuscript is clear and unambiguous. In general, well written with some exceptions.

Some references are missing in the text but in general they are sufficient.

Figures are well presented, although Figure 4 needs correction.

Hypothesis is relevant and well presented.

Experimental design

The research fits well within Aims and Scope of the journal.

Research question well defined, relevant & meaningful.

Investigation was performed rigorously.

Some details are missing in Methods and require to be included. As it is presented, there's no enough information to replicate the study.

Validity of the findings

Novelty state.

Findings and method are meaningful and relevant.

Additional comments

Dear authors,

I have reviewed the manuscript “Predicting in disease occurrence of cabbage Verticillium wilt in monoculture using species distribution modeling” for possible publication in PeerJ. In this study, the authors used species distribution modeling to predict the potential occurrence Verticillium wilt using survey data in central Japan. The study has great value as it is a pioneer at using SDM to predict soil-borne plant disease, a topic of great relevance related to food security and sustainable agriculture. The manuscript is in general well written, with some grammatical issues noted in the attached document, although balance among the different sections is not good, especially in the abstract. In its current form is not ready for acceptance, as there are some issues that need to be addressed before publishing.

A major shortcoming of this study is that it is a case study. While this is not a major problem, it is not mentioned anywhere in the manuscript, and all analyses proceeded as if the study could be replicated. Nor is this issue much discussed in the Discussion. While lack of replication is often unavoidable in SDMs studies at a low spatial scale particularly using survey data, this has to be reported as a limitation of the study.

The abstract is not well balanced. The authors present too much background and just a few details on the results, mentioning mainly how the model was evaluated. Here the authors need to highlight their findings and the relevance of their study.

The introduction reads well and has all the important topics. Although, I suggest adding more information regarding the uses and importance of SDMs. Here, it is important that the authors differentiate between MaxEnt the algorithm and the software.

Material and Methods need more information. For example, authors need to include a justification of the selected variables. A simple description is given for TWI in L157 (A high TWI indicates that a site is moist) but it is not enough. Why the authors selected those variables? How do they influence Verticillium wilt occurrence? Because there is no justification for variable selection, the information on variable contribution feels weak later in the discussion. Also, I am curious to know why no climate predictors were used to model potential occurrence.

In L190, the authors provide the percent contribution of each explanatory variable. I suggest to report instead, or at least include also in the manuscript, the permutation importance of each variable, as this parameter is more relevant and informative within MaxEnt outputs.

L192-193. The authors must justify why they used Maxent's default setting. This point is very important since the authors are basing their results with this configuration without trying to optimize the model. There are many articles that have explored this problem (see Anderson and Gonzalez 2012; Warren and Seifert 2011; Merow et al. 2013; Radosavljevic and Anderson 2013; Syfert et al. 2013; Halvorsen et al. 2016). I would have expected that at the very least and to increase the quality of the models, the ‘hinge’ and ‘threshold’ features would have been deactivated to avoid overfitting response curves.

Regarding the evaluation of model performance, the authors only used the AUC value, which although it is high, does not show much, only that the model is not bad. Getting high values of AUC is very common for models based on presence-only data such as the one developed in this work. If the model is evaluated with cross-validation (for example, using spatial blocks, as in Roberts et al. 2017, http://onlinelibrary.wiley.com/doi/10.1111/ecog.02881/full), a high value of AUC would be more convincing. AUC levels for good performance are only relevant for presence/absence data. Since the authors base all the performance of the model on this statistic, it is necessary to justify it, especially since problems with AUC are well documented (see Lobo et al. 2007). Another option would be to calculate an additional statistic, such as ‘true skill statistics (TSS)’, to corroborate model performance.

How was the 'background' selected for the model? There is no mention of this anywhere in Methods and it is important to describe it.

As the results are just one paragraph, I suggest combining these with the Discussion. Although, I am not sure if the journal accepts a section combining “Results & Discussion”.

In the results, I don’t understand why the authors did not project their model to the whole agricultural area in Tsumagoi village (i.e. all grid cells in Figure 3A, including outside the agricultural area). This would be easy to do and informative considering agricultural expansion. This will provide information on which areas are more vulnerable to the disease in potential new locations.

In L223-224 the authors say "Some grid cells where Verticillium wilt of cabbage had never been recorded had a high probability of disease occurrence"; here, the authors need to discuss these findings. This is very important, is it because these grid cells have the optimal conditions for the occurrence of the disease? This is the place to speculate and discuss in more detail the response of the disease to the selected variables and the conditions across the landscape.

The authors also need to provide a better discussion on the contribution to the prediction for the variables. Specifically, including why TWI (7.7%) and slope (7.1%) had the lowest values. This has to be connected to the justification of the variable selection (see comment above)

There’s no need to have two panels in Figure 4. Just present Figure 4B.

There’s no need to include in figure 3 and 4 legends "These maps in this figure were arranged by ArcMap (ver. 10.2.2, ESRI, Redlands, CA, USA)" and "The occurrence probability calculated in each grid was arranged by ArcMap", respectively.

See more specific comments in the document attached.

Annotated reviews are not available for download in order to protect the identity of reviewers who chose to remain anonymous.

Reviewer 2 ·

Basic reporting

This manuscript describes a substantial research effort to assess potential distribution of cabbage Verticillium wilt in a specific area in Japan. Literature reference is not sufficient, because different sections such as Introduction and Discussion are without a background of the disease, niche theory and on species distribution models. All results are inconsistent, since they were obtained of a weak and unclear modelling approach and a limited data set. A manuscript should show a relevant objective. Species distributions models are not only maps. You need to improve this manuscript a lot with all suggestions included in this document.

Experimental design

I included in a file with commented details of the experimental design.

Validity of the findings

I included in a file with commented details of the validity of the findings.

Annotated reviews are not available for download in order to protect the identity of reviewers who chose to remain anonymous.

Reviewer 3 ·

Basic reporting

Lines 23-24: Please review the use of infected (the host) vs. infested (soil, inanimate objects) here and throughout (Lines 79, 80, . I would tend to write this sentence as “…IPM methods are sometimes inadequate to bring about control in severely infested fields.” since the field is infested, unless you are talking about the plants in the field.
Line 26: do you mean disinfestation?
Line 29: “…to a soilborne…”
Lines 66 and 72: change disinfection to disinfestation
Lines 122-124: I think more information on the pathogens could be provided here (e.g. how long it survives in soil, host range – wide in V .dahliae vs. brassicas in V. longisporum. What is there relative abundance in relation to each other? This can be important depending on what other crops (hosts vs non-hosts) are rotated in the fields.

Experimental design

Lines 112-119: what other crops are grown here? Are other hosts grown in rotation? Does infected seed play a role in the introduction of the pathogen into fields?
Line 121: are certain cultivars of cabbage grown that are resistant or susceptible? Verticillium may go undetected in fields planted to resistant cultivars or detected more frequently in fields planted more often to susceptible cultivars.
Line 125: Were fields sampled more or less equally, or is there any potential bias for certain cells?
Line 135: Do some fields or areas have longer histories of cabbage production than others? Since Verticillium can survive in the soil and buildup over time, would fields with longer histories of cabbage production be identified as infected more instances over time than newer fields?
Lines 157-158: what was the sample size (n)?

Validity of the findings

Lines 77-78: what about exclusion and prevention?
Lines 80-82: it could also be mentioned that knowledge of infested and non-infested fields could help growers prevent infesting new fields or re-introducing the pathogen to fields that have been fumigated.

Additional comments

The manuscript "Predicting disease occurrence of cabbage Verticillium wilt in monoculture using species distribution modeling" by Ikeda and Osawa describes research to predict this important disease in a major production area using historical survey data and novel, ecological-based approaches. Their analyses showed that road density was a major predictor of Verticillium occurrence in this agricultural area. These findings make perfect sense, as the pathogen is soilborne (seedborne in some hosts) and primarily moved with soil or agricultural products via human activities. I think this is a very important and interesting work and recommend publication after the authors address/respond to my comments.

Reviewer 4 ·

Basic reporting

The manuscript is clearly written. There are some relevant references missing.
Please note the following references when you state that there is no previous studies utilizing SDMs in plant pathogen epidemics.
Modeling and mapping the current and future distribution of Pseudomonas syringae pv. actinidiae under climate change in China Wang R, Li Q, He S, Liu Y, Wang M, et al. (2018) Modeling and mapping the current and future distribution of Pseudomonas syringae pv. actinidiae under climate change in China. PLOS ONE 13(2): e0192153. https://doi.org/10.1371/journal.pone.0192153

Narouei-Khandan, H.A., Harmon, C.L., Harmon, P. et al. Potential global and regional geographic distribution of Phomopsis vaccinii on Vaccinium species projected by two species distribution models. Eur J Plant Pathol 148, 919–930 (2017). https://doi.org/10.1007/s10658-017-1146-4

When, where, and whether to manage a plant epidemic
Nik J. Cunniffe, Richard C. Cobb, Ross K. Meentemeyer, David M. Rizzo, Christopher A. Gilligan
Proceedings of the National Academy of Sciences May 2016, 113 (20) 5640-5645; DOI: 10.1073/pnas.1602153113

I also suggest you discuss more on the road network in disease occurrence in the light of other studies eg.
The spread of a wild plant pathogen is driven by the road network
Numminen E, Laine AL (2020) The spread of a wild plant pathogen is driven by the road network. PLOS Computational Biology 16(3): e1007703. https://doi.org/10.1371/journal.pcbi.1007703

Experimental design

The experimental is adequately designed.

Validity of the findings

The authors should discuss how their findings could be extended to wider areas. Is it possible that this model can be used in other Japanese or global areas and why.

---

## Round 0.2 · Minor Revisions

I agree with the two reviewers in that it is necessary an English editorial review, and that the Discussion needs clarity. One of the reviewers included comments in the attached file, please consider them.

Reviewer 1 ·

Basic reporting

I have reviewed the revised manuscript "Predicting disease occurrence of cabbage Verticillium wilt in monoculture using species distribution modeling" for publication in PeerJ.

The authors have made a good effort addressing all my previous comments but the manuscript still has issues that require attention before acceptance.

Experimental design

Some details are required. See comments in the file attached.

Validity of the findings

Some details are required. See comments in the file attached.

Additional comments

Dear authors,
I appreciate your effort addressing all my previous comments; however, the new version of your manuscript has some issues that need to be clarify and corrected before acceptance.

English requires a review, as there are sections that are not easy to follow or understand the ideas.

Annotated reviews are not available for download in order to protect the identity of reviewers who chose to remain anonymous.

Reviewer 3 ·

Basic reporting

line 89 - change to "given"
line 98 - change to "approaches"
line 145 - change to "In recent years, cultivars..."
line 147-148 - change to "...times in total by official and private.....capable of distinguishing..."
line 149 - change to "The surveys and field collections were approved..."
line 173 - change to "roads"
line 258-259 - this sentence is not clear to me.
line 272- change to "uninfested"
line 276 - should "becoming" be changed to "detecting"? This sentence is not clear.
line 278-280 - This sentence is not clear. Do the authors mean that the disease was present but not detected due to some unknown control measures that were being used?
line 285 - delete "more"

Experimental design

Is there a(n) (auto)correlation between road density and agricultural production that would confound these results (i.e. there are more roads in agricultural areas, and more Verticillium in agricultural areas, hence road density and Verticillium appear to be correlated)?

Validity of the findings

No comment

Additional comments

The authors addressed my comments on the first draft. I just have a few more comments/corrections to note.

---

## Round 0.3 · accepted · Accept

I appreciate your effort for considering suggestions by the reviewers in the two rounds of reviews. My ony suggestion is that when you receive the proofs you need to check that all scientific names are in italics.